# Efficacy of Onabotulinum Toxin A on Obsessive–Compulsive Traits in a Population of Chronic Migraine Patients

**DOI:** 10.3390/brainsci12111563

**Published:** 2022-11-17

**Authors:** Giovanna Viticchi, Lorenzo Falsetti, Sergio Salvemini, Marco Bartolini, Silvia Paolucci, Laura Buratti, Mauro Silvestrini

**Affiliations:** 1Neurological Clinic, Marche Polytechnic University, 60020 Ancona, Italy; 2Internal and Subintensive Medicine, Ospedali Riuniti, 60020 Ancona, Italy

**Keywords:** chronic migraine, medication overuse headache, obsessive–compulsive disease, onabotulinum toxin A

## Abstract

Background: Drug addiction may play an important role in chronic migraine (CM) with medication-overuse headache (MOH). Psychiatric diseases are associated with CM, but data regarding obsessive–compulsive disorder (OCD) are lacking. We aimed to establish the prevalence of OCD traits in CM patients with MOH and the impact on onabotulinum toxin A (OBT-A) treatment. Methods: A total of 75 patients with CM and MOH undergoing treatment with OBT-A in our Headache Centre were evaluated. At baseline and after four injection sessions, we assessed the migraine burden and the presence of OCD traits with the Obsessive–Compulsive Inventory—Revised (OCI-R) test. Results: At baseline, 28% of patients had OCI-R scores compatible with borderline OCD aspects, while 22.7% were pathological. An improvement in headache was significantly associated with an increase in the number of subjects with a normal OCI-R score at T0 and T1, whereas patients with a pathological OCI-R score at T0 showed a significantly higher prevalence of CM at T1. Conclusions: Our data showed a significant rate of OCD traits at baseline, which could strengthen the hypothesis of an addictive disorder underlying CM with MOH. OCD traits seem to influence the OBT-A response. An OCD assessment could be useful in improving patients’ selections before starting treatments.

## 1. Introduction

Patients with chronic migraine (CM) represent a serious medical problem with a high rate of treatment failure. Very often, CM is associated with medication-overuse headache (MOH), which significantly increases the complexity of the therapeutic management of patients. Several studies have shown that headaches may be triggered by or associated with several conditions, including hormonal fluctuations [1], circadian rhythm [2], and metabolic alterations [3]. In addition, psychiatric diseases, depression, and anxiety, in particular, are considered typical comorbidities of migraine patients. Compared to non-migraineurs, migraineurs have a 2.2 to 4.0 and 2 to 10-fold increased risk of developing depression and anxiety, respectively [4,5]. In addition, psychiatric illness is a risk factor for the development of chronic episodic migraine [6,7]. Several studies have shown that psychiatric comorbidities are mainly present in CM with MOH [8], and it seems that they may impair the response to treatment by increasing drug cravings [9].

Few studies have explored the presence of obsessive–compulsive disorder (OCD) in migraine patients. The Hadas study showed that OCD was significantly more prevalent in patients with migraine associated with tension-type headaches than in patients with only one form of headache [10]. Curone et al. found a significant proportion of OCD in patients with CM and MOH [11] and formulated the hypothesis that a large proportion of patients with CM suffer from a subclinical form of OCD that may therefore be underdiagnosed and undertreated [12]. Subclinical OCD or OCD personality traits negatively influence the response to standard migraine prophylaxis [13] and more advanced therapies, such as monoclonal antibodies [14]. To our knowledge, no study has explored the impact of onabotulinum toxin A (OBT-A) [15] on CM patients taking into account OCD personality traits.

This study aimed to explore the presence of OCD traits in a group of CM and MOH patients. In addition, we analysed the interactions between obsessive–compulsive scores and OBT-A treatment on patients’ clinical outcomes.

## 2. Materials and Methods

We designed a single-cohort prospective study, considering consecutive subjects attending the Headache Centre of our Neurological Clinic for CM and MOH, undergoing treatment with OBT-A over a two-year period from 1 October 2019 to 30 September 2021. For this study, we chose to include only patients with a follow-up of at least three years prior to the start of therapy to obtain a complete clinical history regarding their headaches. Each subject had a diagnosis of CM and MOH, according to the guidelines of the International Headache Society [16], made by the senior headache researcher of our Headache Centre. During the period of treatment with OBT-A, we allowed each type of medication for acute attacks. 

Before starting OBT-A treatment, each subject underwent a brain imaging evaluation (CT or MRI). Inclusion criteria were: (a) diagnosis of CM with MOH according to international guidelines; (b) clinical indication for OBT-A therapy according to international guidelines; (c) no significant reduction in migraine severity after prophylaxis with at least three classes of drugs among those indicated by international guidelines (beta-blockers, calcium channel blockers, antidepressants, and antiepileptics); (d) no significant reduction in migraine severity after at least one course of steroid detoxification; (e) age > 18 years; (f) discontinuation of all prophylactic drugs after initiation of OBT-A therapy; (h) normal neurological and general examination at baseline and follow-up visits.

Exclusion criteria were: (a) regular intake of drugs acting on the nervous system (e.g., antiepileptics) except for drugs used for migraine prophylaxis; (b) regular intake of drugs acting on the mood system (neuroleptics, antidepressants, anxiolytics) except for drugs used for migraine prophylaxis; (c) history of psychiatric diseases; (d) irregular intake of or intolerance to OBT-A therapy; (e) brain imaging (CT or MRI) showing the presence of tumours, ischaemic or haemorrhagic lesions, or other significant brain changes; (f) presence of current or past neurological disease, including trauma, vascular incidents, or exposure to toxic substances. 

At the time of the first visit to start treatment with OBT-A (T0), each patient was assessed by means of the MIDAS test and the HIT-6 test to determine their headache burden. All patients were then asked to complete the following questionnaires: the Barratt Impulsiveness Scale (BIS-11), the Psychopathic Personality Inventory (PPI), and the psychometric test of the Brief Resilience Scale (BRS). To assess the presence and severity of OCD personality traits, we used the validated Italian version of the Obsessive–Compulsive Inventory—Revised (OCI-R) scale [17]. The OCI-R is a self-report scale that assesses the severity and type of symptoms consistent with OCD. We decided to consider only the total score and not the individual domains due to the low sample size.

Each patient had an OBT-A treatment session every three months: for this study, we evaluated the responses to OBT-A therapy over a year of treatment (a total of four injection sessions). According to the PREEMPT study protocol, in each session, we performed 31 local injections with 5 I.U. of OBT-A for a total of 155 I.U. [18]. In selected cases, we increased the dosage of OBT-A up to a maximum of 195 I.U. [19].

Patients regularly completed a headache diary to monitor the severity, intensity, and frequency of their migraines. During the last administration session (T1), we submitted each patient to the same battery of questionnaires. 

All participants gave written informed consent according to the Declaration of Helsinki. The Ethics Committee of the Marche Region (CERM), Italy, approved the study (protocol number 202266).

### Statistical Analysis

We collected, as continuous variables: age, MIDAS score (at T0 and T1), number of migraine attacks (at T0 and T1), BS-11 score (at T0 and T1), PPI score (at T0 and T1), BRS-6 score (at T0 and T1), and HIT-6 score (at T0 and T1). The OCI-R score (at T0 and T1) was analysed as a categorical variable, considering four discrete states based on the scores obtained: normal, borderline, psychological distress, and pathological; we also treated and compared the same variables as continuous.

We summarised the following variables as dichotomous: gender, use of medication prophylaxis for a migraine attack, use of medication for an acute attack, and clinical improvement after OBT-A treatment. We then recoded the migraine typology based on the number of attacks at T0 and T1 into two new binary variables, defining “episodic” as a migraine characterised by <15 attacks per month and “chronic” as a migraine characterised by ≥15 attacks per month. We also dichotomised the OCI-R results into two new variables defined by “non-pathological” (normal, borderline, psychological distress) and “pathological” values. Patient data were collected anonymously in an MS Excel file, which was then transformed into an SPSS file used for data analysis.

We tested continuous variables for normality using the Kolmogorov–Smirnov test. Normally distributed variables were presented as mean and standard deviation (SD) and compared with the t-test for independent or paired samples; non-normally distributed variables were presented as a mean and interquartile range [IQR] and compared with the Mann–Whitney U test (independent variables) or the Wilcoxon signed-rank test (paired variables). Categorical and dichotomous variables were presented as numbers and percentages and compared with the chi-square test. We considered a “responder” to be any subject with a reduction in the number of seizures or a reduction in the MIDAS score of at least 50% [19,20].

We chose, as a multivariate model, a generalised linear model (GLM) for repeated measures considering (i) the repeated measure of interest as the dependent variable, (ii) clinical improvement as the main predictor, and (iii) age, sex, prophylaxis, and treatment of migraine attacks as covariates. The statistical analysis was conducted with SPSS 13.0 for Windows Systems (SPSS Inc., Chicago, IL, USA).

## 3. Results

### Sample Enrolled

We selected 112 consecutive CM patients with MOH undergoing OBT-A therapy. Among them, we excluded a total of 37 patients: 26 reported a common intake of drugs influencing mood, 3 had a previous diagnosis of psychiatric disease, 4 did not complete the four-session course of OBT-A therapy, and 4 had significant changes in brain imaging. We obtained a final sample of 75 subjects. The baseline characteristics and comparisons between T0 and T1 are summarised in Table 1. Of note, while all patients had chronic migraine at T0, at T1, 42 subjects (56.0%) were reclassified as affected by episodic migraine. We did not find significant differences between the patients submitted to 195 I.U. doses of OBT-A and the patients treated with doses of 155 I.U.

Notably, after treatment, a significant proportion of subjects had fewer than 15 migraine attacks per month, as shown in Figure 1, and a reduction in MIDAS score. In addition, we observed a significant reduction in the median OCI-R between the two time points, as shown in Table 1.

A reduction in headache frequency was significantly more common in subjects with a normal OCI-R at T0 and T1, as shown in Table 2 and Figure 2.

In addition, patients with a pathological OCI-R at T0 showed a significantly higher prevalence of chronic migraine at T1 (non-pathological OCI-R at T0: 22 patients (29.3%); pathological OCI-R at T0: 11 patients (14.7%); *p* = 0.051). 

The GLM/Repeated Measures Model considered pre- and post-treatment OCI-R as the main dependent variable, clinical improvement as the main independent variable, and age, sex, pharmacological prophylaxis, and treatment of the attack as covariates. We observed that an improvement in headache symptoms was significantly associated with a reduction in the mean OCI-R value (T0: 1.08; 95% CI: 0.75–1.42; T1: 0.59; 95% CI: 0.279–0.893; *p* = 0.001), while the absence of improvement was associated with a non-significant reduction in the mean OCI-R value (T0: 1.55; 95% CI: 1.11–1.99; T1: 1.44; 95% CI:1.03–1.85; *p* = 0.288), as shown in Figure 3.

## 4. Discussion

Our data suggest that OCD traits may exert a complex influence on CM patients and affect the effect of OBT-A treatment. We noted that at baseline assessment, a large proportion of patients had scores indicating OCD traits. According to the OCI-R scale, 28% of patients at T0 had scores compatible with borderline aspects, while 22.7% of the sample had an OCI-R score indicative of a pathological diagnosis. 

This result confirms the hypothesis that patients with CM may often have a subclinical and unrecognised form of OCD [12], with a possible relevant influence on treatment efficacy. 

The possible causes of both CM and MOH have been extensively investigated but not completely established. Some investigations have found an association between MOH and abuse behaviour [21], highlighting patients’ addiction-like behaviours and tendency to overuse opioids [22]. People with MOH often present with a lack of control over impulsivity, which could partly explain the evolution from an episodic to a chronic form with drug abuse [12]. All of these elements are also common in OCD and may justify the high proportion of these traits in CM patients. On the other hand, as suggested by several studies, MOH and OCD share the tendency for compulsive substance abuse to counteract anxiety and fear, which, in the case of migraineurs, is linked to the expectation of the next headache attack [12].

An attractive hypothesis is that compulsion towards medication use may be the key element in the development of MOH. Several studies have shown that patients with MOH often attain such significant abuse behaviour that they are defined as ”dependent” on medication for acute headache attacks according to the Diagnostic and Statistical Manual of Mental Disorders (DSM)-IV criteria [23]. MOH and OCD seem to share a common pathophysiological substrate because both appear to present alterations in the striatal—thalamic—orbitofrontal circuit [8,24,25]. Recent studies have shown that compulsive drug addiction behaviours and MOH share similar cognitive problems, including the impairment of decision-making mechanisms and an imbalance of the adaptive reward systems [26]. Functional imaging studies have shown alterations in the mesocorticolimbic reward circuitry in both conditions [26,27]. 

In our study, by analysing different classes of OCR-R scores, we noted that patients with normal or borderline profiles evolved towards an overall improvement in both migraines and psychological traits. On the other hand, patients with pathological scores did not show a positive evolution. Consequently, the presence of pathological OCD traits, probably due to the extremely in-depth structuring of the abuse behaviour, seems to negatively influence the evolution of the migraine burden, despite a specific treatment such as OBT-A.

On the other hand, patients with an improvement in headache after OBT-A also presented an improvement in their OCD-R score. Similar results were described for scales assessing addiction-like behaviour [22]. OBT-A seems to have a positive effect on both migraine severity and mild forms of OCD. Patients with borderline profiles switched to a lower severity class of OCD-R after only one year of treatment. Some studies have pointed to an impact on depressive symptoms after treatment with OBT-A [28,29]. A recent investigation by Blumenfeld et al. demonstrated a positive impact of OBT-A therapy on depression and anxiety symptoms, as well as an improvement in sleep quality and fatigue [30]. 

To our knowledge, this is the first study to show the effect of OBT-A therapy on OCD. The pathophysiological basis may be the action on the limbic system. Onabotulinum toxin A has a peripheral effect of modulating proprioceptive inputs and results in a significant effect on the limbic system due to the trigeminal nucleus lowering the sensory afferents’ inputs [30]. Furthermore, a derangement of the mesolimbic reward system, in particular the mesocorticolimbic dopaminergic circuit, has been described in OCD patients. This is similar to that which has been documented in addicted individuals [26]. These data have also been confirmed by functional MRI connectivity studies [31].

Another possible mechanism to explain the improvement in OCD during OBT-A treatment is the reduction in drug abuse related to the reduction in the number of headache attacks. By reducing drug intake, patients tend to reduce their anxiety and compulsivity and, eventually, improve their reward mechanisms. In our sample, people with improved OCD showed a significant reduction in their MIDAS score, which expresses the impact of migraine on patients’ quality of life.

OBT-A is a safe, well-tolerated, and highly CM-specific therapy. Based on the results of our study, we hypothesise that patients with subclinical or mild OCD are likely to benefit from OBT-A treatment, whereas the effectiveness of this therapy is not guaranteed for patients with pathological OCD scores. 

This study presents some limitations. The first limitation of this study is the small sample size. While it is planned to enlarge our sample to obtain more robust data in the future, it is important to stress that OBT-A therapy is dedicated to a limited sample of subjects resistant to previous drug prophylaxis. 

Another limitation is the lack of a control group. This is a difficult problem to solve due to the ethical considerations related to the need to offer treatment to all patients with MOH who do not show a response to steroid detoxification. 

## 5. Conclusions

OBT-A seems to effectively reduce the tendency for addiction and drug dependence in CM patients with MOH. The most obvious explanation seems to be related to the reduced frequency of attacks and, consequently, the need to take medication. Based on our results, patient selection may play a central role in achieving a positive response to treatment. Patients with CM and psychiatric comorbidities, including OCD, seem to have a reduced benefit from specific therapies such as OBT-A, as already demonstrated for monoclonal antibodies targeting the C-GRP pathway [14]. 

Finally, for patients with mild OCD, OBT-A treatment appears to be effective, whereas, for severe forms of OCD, the approach is probably not sufficient. In this light, treatment with OBT-A in patients with severe OCD should probably be preceded by a psychiatric evaluation and psychological support therapy.

## Figures and Tables

**Figure 1 brainsci-12-01563-f001:**
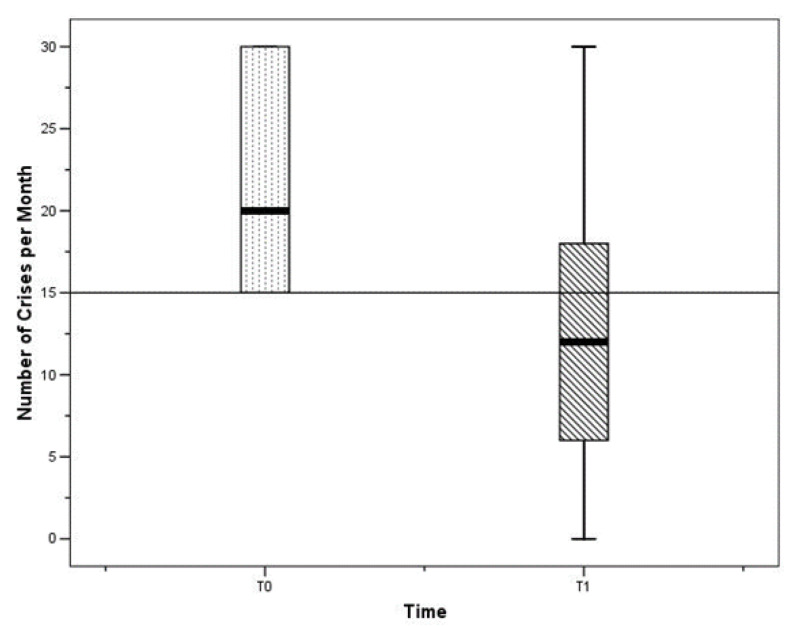
Differences between T0 and T1 in the monthly number of crises.

**Figure 2 brainsci-12-01563-f002:**
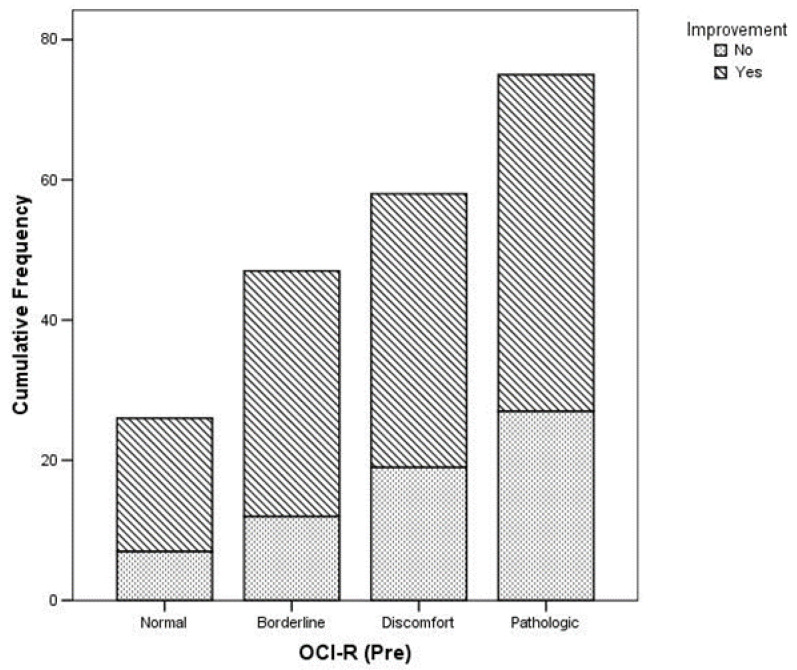
Relationship between OCI-R at T0 and clinical improvement (*p* = 0.002).

**Figure 3 brainsci-12-01563-f003:**
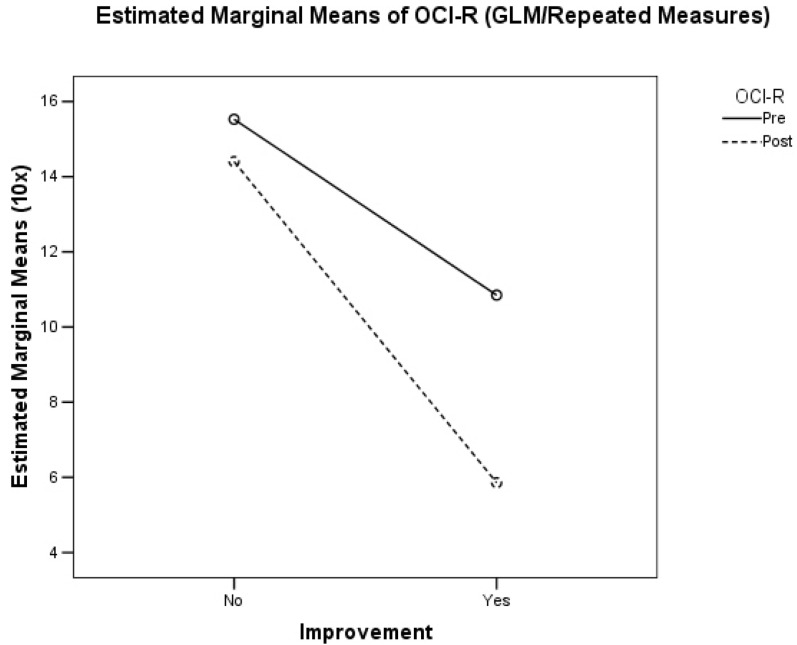
GLM/Repeated Measures Model results (*p* < 0.0001). The Estimated Marginal Means were magnified 10 times to improve the readability of the Figure.

**Table 1 brainsci-12-01563-t001:** Baseline characteristics of the cohort at T0 and T1.

	T0	T1	*p*
Age	50.8 (±11.1)	--	--
Female Sex, (n, %)	55 (70.5%)	--	--
Clinical Improvement, (n, %)	--	48 (61.5%)	--
MIDAS Score (±SD)	77.43 (±52.8)	41.16 (±39.4)	0.0001
Crises per Month (±SD)	21.51 (±6.59)	12.65 (±8.31)	0.001
Migraine Type:			
Chronic (n, %) Episodic (n, %)	75 (100%) 0 (0.0%)	33 (44.0%) 42 (56.0%)	0.0001
BS-11 (±SD)	9.17 (±1.36)	7.53 (±2.02)	0.0001
PPI (±SD)	4.16 (±0.81)	3.17 (±1.15)	0.0001
BRS-6 (±SD)	4.36 (±0.75)	3.39 (±1.25)	0.0001
HIT-6 (±SD)	18.04 (±2.99)	14.67 (±4.06)	0.0001
Drug Prophylaxis at T0 (n, %)			
Topiramate Valproic Acid Amitriptyline Propranolol Gabapentin Lamotrigin Pregabalin No therapies	20 (26.7%) 5 (6.7%) 33 (44%) 4 (5.3%) 5 (6.4%) 1 (1.3%) 2 (2.7%) 5 (6.67%)	--	--
Drug Treatment of Attacks (n,%)			
NSAIDS Triptans Indomethacin Paracetamol	7 (9.3%) 35 (46.67%) 11 (13.3%) 3 (4.0%)	--	--
OCI-R (IQR)	1 (2)	0 (2)	0.0001
OCI-R			
Normal Borderline Discomfort Pathologic	26 (34.7%) 21 (28.0%) 11 (14.7%) 17 (22.7%)	42 (56.0%) 11 (14.7%) 10 (13.3%) 12 (16.0%)	0.0001

**Table 2 brainsci-12-01563-t002:** Differences between responder and non-responder subjects.

	Responder	Non-Responder	*p*
MIDAS Score at T0 (IQR)	70 (84)	55 (36)	0.233
MIDAS Score at T1 (IQR(	20 (20)	58 (61)	0.0001
Number of Crises at T0 (IQR)	18.5 (15)	20 (15)	0.871
Number of Crises at T1 (IQR)		20 (12)	0.871
OCI-R at T0 (IQR)	2 (3)	1 (2)	0.065
OCI-R at T1 (IQR)	2 (3)	0 (1)	0.001
OCI-R at T0 (n, %)			
Normal Borderline Discomfort Pathologic	19 (39.6%) 16 (28.0%) 4 (8.3%) 9 (18.8%)	7 (25.9%) 5 (14.7%) 10 (13.3%) 12 (16.0%)	0.0001
OCI-R at T1 (n, %)			
Normal Borderline Discomfort Pathologic	33 (44.0%) 8 (10.7%) 2 (2.7%) 5 (6.7%)	9 (12.0%) 3 (4.0%) 8 (10.7%) 7 (9.3%)	0.002
BS-11 at T0 (±SD)	9.33 (±0.93)	8.89 (±1.89)	0.176
BS-11 at T1 (±SD)	6.79 (±1.81)	8.85 (±1.68)	0.0001
PPI at T0 (±SD)	4.25 (±0.73)	4.00 (±0.92)	0.199
PPI at T1 (±SD)	2.77 (±1.08)	3.89 (±0.93)	0.0001
BRS-6 at T0 (±SD)	4.42 (±0.71)	4.26 (±0.81)	0.385
BRS-6 at T1 (±SD)	2.90 (±1.19)	4.26 (±0.81)	0.0001
HIT-6 at T0 (±SD)	18.44 (±3.29)	17.33 (±2.27)	0.126
HIT-6 at T1 (±SD)	13.48 (±4.24)	16.78 (±2.69)	0.0001

## Data Availability

Data are available on request due to restrictions (e.g., privacy or ethical).

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
