# Peer review of "Efficacy of Onabotulinum Toxin A on Obsessive–Compulsive Traits in a Population of Chronic Migraine Patients"

_brainsci, 2022, doi:10.3390/brainsci12111563_

Round 1
Reviewer 1 Report
Dear authors
, thank you for your efforts. I have listed my
corrections which is marked with similar colors in the text
below.There are inconsistencies between the inclusion and
exclusion criterias (line numbers: 72, 73, 74, 79, 80, 81, 82) :
f- Discontinuation of all prophylaxis drugs is an inclusion
criterion and also an exclusion criterion???
h- normal neurological examination is an inclusion
criterion and also an exclusion criterion??
There is also a duplication in the exclusion criteriaThere is no control group, as there are some deficiencies, but in fact, it is emphasized that ethical problems cannot be made and one of the applied ones is highlighted,
Author Response
Dear Reviewer,
We were pleased to have the opportunity to revise our paper and in the revised version we have carefully considered your suggestions. As instructed, we have attempted to succinctly explain changes made in response to all comments in a point-by point fashion [Query(Q)/Answer(A)].
Dear authors, thank you for your efforts.
I have listed my corrections which is marked with similar colors in the text below. There are inconsistencies between the inclusion and exclusion criterias (line numbers: 72, 73, 74, 79, 80, 81, 82):
Qf- Discontinuation of all prophylaxis drugs is an inclusion criterion and also an exclusion criterion???
Af: We apologized for this inaccuracy. “Discontinuation of all prophylaxis drugs” is an inclusion criterion, and in the revised version we deleted this sentence from the exclusion criteria (line numbers 80-81).
Qh- normal neurological examination is an inclusion criterion and also an exclusion criterion??
Ah: We apologized for this inaccuracy. “Normal neurological examination” is an inclusion criterion, and in the revised version we deleted this sentence from the exclusion criteria (line number 81)
Q: There is also a duplication in the exclusion criteria
A: We agree with the reviewer. “brain imaging (CT or MRI) showing evidence of neurological disorders” and “brain imaging (CT or MRI) showing the presence of tumours, ischaemic or haemorrhagic lesions or other significant brain changes” are a duplication of the same exclusion criteria. In the revised version we delete the duplication (line number 79-80).
Q: There is no control group, as there are some deficiencies, but in fact, it is emphasized that ethical problems cannot be made and one of the applied ones is highlighted,
A: We thank the reviewer for this specification. The lack of a control group is one of the limits of the study, but, as specified in the Limitations section, we could not deny effective treatment for MOH to some patients.
Reviewer 2 Report
The paper presented to me for review is "Efficacy of Onabotolinum toxin A on obsessive-compulsive 2 traits in a population of chronic migraine patients." Although the efficacy of Botox in the treatment of chronic migraine is proven and it is known to have a particularly positive effect on the reduction of allodynia and hypersensitivity there is still a lack of real-life studies regarding the effect of the toxin on other disease aspects in migraine patients. The authors in 75 patients with CM and MOH conducted treatment with OBT-A also evaluating obsessive-compulsive disorder (OCD) during treatment. The issue is new.
The paper is written very coherently and well planned and the graphical representation of the results is correct. The references are mostly new and appropriate.
Minor corrections to be considered before accepting the paper for publication:
1. in the exclusion criteria, it should be clearly emphasized that in addition to normal imaging findings and a normal neurological examination, the patients had no current or past neurological disease including trauma, vascular incidents or exposure to toxic substances
2. whether the use of a dose of 155j. vs. 195j. onabotolinum toxin A was associated with even greater improvement in patients?
Author Response
Dear Reviewer,
We were pleased to have the opportunity to revise our paper and in the revised version we have carefully considered your suggestions. As instructed, we have attempted to succinctly explain changes made in response to all comments in a point-by point fashion [Query(Q)/Answer(A)].
The paper presented to me for review is "Efficacy of Onabotolinum toxin A on obsessive-compulsive 2 traits in a population of chronic migraine patients." Although the efficacy of Botox in the treatment of chronic migraine is proven and it is known to have a particularly positive effect on the reduction of allodynia and hypersensitivity there is still a lack of real-life studies regarding the effect of the toxin on other disease aspects in migraine patients. The authors in 75 patients with CM and MOH conducted treatment with OBT-A also evaluating obsessive-compulsive disorder (OCD) during treatment. The issue is new.
The paper is written very coherently and well planned and the graphical representation of the results is correct. The references are mostly new and appropriate.
Minor corrections to be considered before accepting the paper for publication:
Q1. in the exclusion criteria, it should be clearly emphasized that in addition to normal imaging findings and a normal neurological examination, the patients had no current or past neurological disease including trauma, vascular incidents or exposure to toxic substances
A1: We thank the reviewer for this correct suggestion. In the revised version we add an exclusion criterion according to the reviewer’s indication (line numbers 83-84).
Q2. whether the use of a dose of 155j. vs. 195j. onabotolinum toxin A was associated with even greater improvement in patients?
A2: We thank the reviewer for this suggestion. We did not find significant differences about the migraine improvement in patients submitted to 195j dose on respect to 155j. In the revised version we better specified this point (Results section, line numbers 144-145).